# Close Companions? A Zooarchaeological Study of the Human–Cattle Relationship in Medieval England

**DOI:** 10.3390/ani11041174

**Published:** 2021-04-20

**Authors:** Matilda Holmes, Helena Hamerow, Richard Thomas

**Affiliations:** 1School of Archaeology and Ancient History, University of Leicester, Leicester LE1 7RH, UK; rmt12@leicester.ac.uk; 2School of Archaeology, Oxford University, Oxford OX1 2JD, UK; helena.hamerow@arch.ox.ac.uk

**Keywords:** cow, oxen, dairy, pathology, draught, human-animal relationships, livestock, social zooarchaeology, ethnography

## Abstract

**Simple Summary:**

The population of medieval England (AD 400–1400) was largely employed in farming-related activities. Cattle were crucial as providers of power as well as milk, meat, and hides and were valued economically and socially. From the mid-seventh century, cattle husbandry increasingly relied on draught cattle for arable production and agricultural tasks such as ploughing, hauling, and carting. Analysis of cattle bones from archaeological sites permits the reconstruction of herd demographics and assessment of the use of cattle for traction through analysis of age and sex profiles, and the presence and severity of pathological and sub-pathological changes to the lower limb bones of cattle. When combined with ethnographic studies and historical documents, it is possible to perceive how attitudes to cattle have changed over time. By integrating multiple lines of evidence (archaeological, ethnographic, and historical), this study reveals how the value of cattle changed over time from a status symbol (representing accumulated wealth) to a commodity. A peak in the use of cattle for traction between the mid-ninth and mid-eleventh centuries may have increased the proximity of human–cattle bonds, which perhaps diminished in subsequent years as the demand for younger cattle increased to feed a growing urban population.

**Abstract:**

Across medieval Europe, cattle commanded a major, if shifting, economic and social value, and their use for meat, milk, and traction is well established. Although the changing roles of cattle throughout this period may have influenced relationships between humans and cattle, this has been largely neglected in historical and zooarchaeological studies. Data from nearly 700 archaeological assemblages of animal remains have been used to provide an overview of the herd structures (age and sex) of cattle populations for England between AD 450 and 1400. These have been analysed alongside pathological and sub-pathological changes in over 2800 lower limb bones of cattle from seventeen archaeological sites to provide a better understanding of the use of cattle for ploughing, hauling, and carting. The findings were considered alongside historical documents and ethnographic evidence to chart changing human–cattle relationships. Results indicate that human–cattle relations varied with changing economic, agricultural, and social practices. From the mid-fifth century, cattle were a form of portable wealth, however, by the mid-ninth century, they were perceived as a commodity with monetary value. From this period, close human–cattle bonds are likely to have been widespread between plough hands and working animals. Such bonds are may have diminished with the increasing number of young beef cattle kept to supply the urban population from the mid-eleventh century.

## 1. Introduction

In medieval England (AD 450–1550), livestock were vital providers of power, fertilizer, and raw materials (such as wool, hair, horn, bone, skin and antler) as well as meat and milk. Cattle were integral to arable production and agriculture as the tractors of their day [1] (p. 222), providing the power to pull ploughs, carts, and other loads. Throughout this period, the majority of the population would have lived in rural locations and been directly involved in farming [2] (p. 393).

The early part of the medieval period was characterized by small, self-sufficient farms, producing enough to feed an extended family alongside a small surplus to provide tribute to the local king or lord, and as insurance against unforeseen circumstances [3,4,5,6]. A few cattle would have been kept by each farmer to milk, pull an ard, and ultimately provide meat security. From around the mid-seventh century, a growing and increasingly urban population as well as a more defined social hierarchy required increased arable output; from the twelfth century, this growth accelerated, resulting in transformations in urban markets, and the re-organization of the rural population in some areas into villages to enable more efficient and centralized production [7,8,9]. As a result, cattle were required to produce more, either directly in the form of primary products of meat and raw materials, or indirectly as secondary products such as traction or milk [4] (p. 128), [10] (p. 213), [11] (p. 118), [12] (p. 37).

Zooarchaeology can be used to investigate the role of cattle within the economy in several ways. Age data provide an estimate of the proportion of animals culled early for meat or later after providing years of secondary products. Sex data can refine the likelihood that secondary products involved a focus on dairying (more cows), and traction can be identified using palaeopathology. When put under mechanical stresses such as those resulting from draught activities, cattle lower leg bones (metapodials and phalanges) adapt through remodelling. Specifically, remodelling includes extra bone formation such as osteophytes and enthesophytes (collectively exostoses), and contour changes of articular surfaces. Correlations therefore exist between changes in cattle foot bones and the use of animals for hauling, carting, and ploughing (e.g., [13,14,15]).

In medieval England, most people would have lived and worked alongside animals every day, which contrasts considerably with the distance between consumers and the source of animal food providers in modern society, where most of the population has no contact with the livestock relied upon to produce meat, milk, and eggs. Thus, differences in the animal experiences of modern and medieval populations makes understanding how they were perceived and cared for problematic. For people that live and work with domesticated animals every day, those animals will be integral to their world view, which is something lost to many urban societies [16] (p. 41), [17] (p. 83). It therefore becomes necessary to draw upon the lived experiences of those who work with cattle as well as historical evidence to understand potential relationships between people and cattle in the past. Ethnographers and social anthropologists have increasingly recognized that human cultures cannot be studied in isolation, but must be integrated with the landscapes and nonhuman animals that surround them [18]. Rather than placing humans and animals in opposition, in separate spheres within the world, a consideration of how farmers worked *with* their animals is vital [19] (p. 76). An increasing number of studies considering past human–animal relationships have been added to the zooarchaeological canon in recent years (e.g., [20,21,22,23]). This has shifted emphasis away from reconstructing production and consumption towards understanding the interactions between people, animals, and their surroundings. However, few studies exist regarding the social interactions between livestock and people in the medieval period [24]. It is vital to attempt a better understanding of the relationships that existed in the past between people and the animals they lived and worked alongside, to provide a meaningful reconstruction of past societies [20] (p. 395). Animals are not passive objects, but active agents that affect how people relate to them, even as changing economic, social, and cultural landscapes lead to transformations in interactions [25,26]. For example, cats are common companion animals in modern homes, and although this remains an unequal relationship [27] (chapter 8), they are often treated like family members, mourned, and buried after death, yet this has not always been the case. The ninth-century poem of Pangur Ban relates the fondness of an Irish scholar for his cat [28], but throughout the medieval period, they were often the subject of cruel sports, and even ‘private cats of the fireside or hearth’ had a price for their skins [29,30] (p. 20).

The aim of this study is to combine zooarchaeological, historical, and ethnographic data to better understand the human–cattle relationship in medieval England. Cattle have been chosen as the focus, given their abundance and importance in the medieval economy. Zooarchaeology is fundamental to understanding human–animal relationships as it provides direct evidence of past animal populations. Zooarchaeological data alone, however, can only provide limited insights into the past treatment of animals relating to their physical remains. Evidence from historical sources is vital to place the findings into an economic, social, and cultural framework, and comparisons with contemporary ethnographic findings regarding human–animal relationships allow a better understanding of the likely connections between animals and people. The following research questions will be addressed:How were cattle used in medieval England, and how did this change through time?What do historical documents imply regarding the value, use of, and attitudes towards cattle in medieval England?Can ethnographic studies of comparable human–animal relationships aid the understanding of how people and cattle co-existed in medieval England?

## 2. Materials and Methods

A recent project tracing the expansion of cereal farming in early medieval England, entitled Feeding Anglo-Saxon England (FeedSax), takes a multi-disciplinary approach, combining data from zooarchaeology, palynology, archaeobotany, isotope analysis, a program of C14 dating, settlement archaeology, and historical sources [31,32]. The zooarchaeological component provided data that will be used as the basis for this investigation. Two strands of data were compiled. The first was a synthesis of published zooarchaeological data from 454 English medieval settlement sites dated to between ca. AD 450 and ca. AD 1400 (Figure 1, Table 1), resulting in 582 phased assemblages (a single site may have data from more than one phase of occupation). A database of all sites included is available from the Archaeology Data Service [33]. Second, 17 sites with large animal bone assemblages of known date were selected and re-analysed to record age, sex, and pathology (Table 2). Broad date ranges were used for analysis, based on commonly accepted medieval periods (early Saxon AD 450–650; middle Saxon AD 650–850; late Saxon AD 850–1066; high medieval AD 1066–1250; later medieval AD 1250–1400).

The synthesis of data from existing site reports (Table 1) included site details (name, location, date, geology, elevation), number of cattle, sheep/goat, and pig bones and teeth and the age and sex profiles of cattle based on tooth wear [34,35], and metric data from sexually dimorphic bones [36,37,38]. Primary data were recorded in detail from targeted sites including age of death from mandibles, sex from metacarpals and pelves, and a suite of pathological and sub-pathological changes in the autopodia.

Simplified mortality profiles are provided based on the likely use of cattle present in each assemblage. If the data suggest that most animals were culled before becoming fully mature, they were classified as resulting from a *meat* strategy. If most were older adults or elderly, they were considered to have been important for *secondary* products (milk or traction). If cattle were culled at a range of age groups, they were recorded as having been from a *mixed* strategy including animals kept for both meat and secondary products.

Sub-pathological and pathological changes were recorded on phalanges and metapodials using an age-independent modified pathological index (mPI) following existing conventions [14,15,39], where a bone exhibiting no change has a mPI of 0, and one with the maximum score has a mPI of 1. Figure 2 illustrates some of the most severe examples of exostoses and lipping, but other recorded characters include eburnation, broadening of articulations, and plantar depressions. The presence and severity of these changes were assessed within each assemblage, with fore- and hind limb elements separated, as the natural discrepancy in weight distribution means that the scores will be greater in the fore limb [15,40]. Prevalence was calculated as the proportion of cattle bones in each phase exhibiting a mPI of more than 0, while severity was calculated as the proportion of elements with a mPI score over 0.4. This cut off was chosen based on a study of draught oxen, which produced a mean of 0.4 using a similar methodology with Table 18 in reference [14].

Few historical documents exist in the early part of the period, being largely limited to the sixth-century writings of Gildas, Bede’s Ecclesiastical History (c. 731), and the Anglo-Saxon Chronicle (c.871–899), none of which specifically refer to cattle. From the tenth century, an increasing body of law codes and charters is available that permit the social and economic value of cattle to be gauged, and these will be used in this study. The earliest medieval text specifically referring to the management of farm animals was the *Le Dite de Hosebondrie* by Walter of Henley, which was published around 1280 and continued to be used well into the sixteenth century [41] (p. 76). Medieval historical documents describing animal husbandry were widely founded on classical texts, information being repeated and recycled into the eighteenth century [42] (p. 73), [43] (p. 17), [44] (p. 235).

Ethnographic studies were incorporated from modern sources describing human–cattle relationships. Studies were chosen that were most likely to reflect the farming economy of medieval England, incorporating non-intensive methods and the use of draught cattle, or to provide insights into the relationships that may form between farm workers and cattle. These range from observations made on small holdings and larger farms in the United Kingdom to communities that rely heavily on the use of cattle for power [45,46,47,48,49,50,51]. The incorporation of observations from social anthropology and ethnography with archaeological research has a long history [52,53] and contributes to an experiential understanding of cattle exploitation and human–cattle relationships in the medieval period by bringing real subjects in familiar situations into the study. However, it is not without challenges, and problems include observer bias, historical and cultural specificity, and a reliance on analogy [27] (p. 60), [52] (p. 402), [54], but by using the method with caution, the potential benefits of insights drawn from populations living and working with cattle are many [55] (p. 1). Medieval cattle husbandry would have required the same basic considerations of cattle welfare and principals of production, be it meat, milk or traction, as those using cattle for the same purposes today [51] (p. 14).

## 3. Results and Discussion

### 3.1. The Zooarchaeology of the Medieval Economy

The results show that cattle in the mid-fifth to mid-seventh centuries were largely used for a mixture of meat and secondary production, with few sites exhibiting specialized production of either meat or secondary products (Figure 3). This is consistent with the largely self-sufficient nature of farming during this period. A change in the economic use of cattle can be observed after the mid-seventh century, when they became more important for secondary products, a trend that peaked in the mid-ninth to eleventh centuries when the majority of cattle populations were culled as old adults or elderly animals after providing many years of milk or traction. However, an increase in the number of assemblages with a mixed strategy can be observed from the mid-eleventh century, increasing further in the mid-thirteenth century. This is consistent with divergent husbandry practices, split between a demand for younger meat animals, the growing importance of the dairy industry, and the use of horses for traction [11,56,57,58]. Attention must be drawn to the very small sample size for the mid-thirteenth to fifteenth century sample; only three assemblages were available, which is too small to interpret reliably, although the exploitation of cattle for meat and milk reflects established trends for the period [58].

A greater proportion of female cattle was observed in all phases (Figure 4), which is consistent with a choice to reduce the number of adult male animals in a herd, given the multi-purpose potential of females. While the very high proportion of female cattle in the mid-fifth to mid-seventh centuries could imply an emphasis on dairy production, given the small-scale, self-sufficient nature of the economy at this time [8] (p. 269), it is more likely that they were kept because of their value in producing calves and milk while also being capable of draught work. An increase in males from the mid-seventh century indicates that from this period, animals were increasingly required for draught work, as males have little value apart from power [51] (p. 15). It is further likely that the majority of these males were oxen (castrated male cattle), as the temperamental nature of bulls make it more likely that males kept into adulthood for draught work were castrated [51] (p. 15), [59] (p. 478). Historical references to oxen exist from the classical writings of Columella (*De re rustica* ch.6, 26, 1), and from one of the earliest examples of an Anglo-Saxon charter, a late eighth century grant of land by Offa [60] (document 78.2.3).

The proportion of cattle affected by pathological and sub-pathological changes to the bones of the feet has altered relatively little through time (Figure 5). Small peaks can be observed in the mid-fifth to mid-seventh centuries, and again in the ninth to eleventh centuries. These are not unexpected in the later period, given the increased use of cattle for secondary products (Figure 3). The peak in the earlier period implies some continuity of Roman husbandry practices, focused on arable production [15]. Severe lesions are relatively rare (Figure 6), with fewer than 15% of bones in any phase exhibiting mPI scores of over 0.40. In general, the combination of a high proportion of the cattle population exhibiting pathological or sub-pathological changes with a relatively low proportion with very severe lesions suggests that although many were used for low-level draught-related work, only a few were used consistently and repeatedly for specific draught purposes over a long enough period of time to cause severe changes to the bones of their lower limbs. The development of changes to the bone resulting from increased stress is highly variable between individuals [14] (p. 62), [61] (p. 129), but some indication of what constitutes the duration of an animal’s working life comes from Bartosiewicz et al.’s study of draught oxen of known age [14]. Working animals aged eight or over had a pathological index of at least 0.32, while the two six-year-old animals were less affected, with scores of 0.17 and 0.22, and the young two-year-old beef bulls had very low scores of 0.01–0.11 Table 18 in reference [14]. Animals with high scores are therefore likely to have been used for traction for several years.

As noted above, the fore limb elements will exhibit higher pathological index values than hind-limb elements because a greater proportion of the animal’s body weight is supported by the front legs [14] (p. 61). Therefore, the greater severity of mPI values observed in hind-limb elements from the mid-seventh century implies that the hind limbs were subject to greater loading than the fore limbs. As this coincides with data indicating an increase in both secondary products and male cattle (Figure 3 and Figure 4), it is reasonable to suggest that this reflects a greater emphasis on draught use. While this increase in traction is consistent with the need for greater agricultural output over time, the peak in severity in the mid-ninth to mid-eleventh centuries combined with a peak in evidence for secondary production, implies that this was a period of considerable change in the cattle economy.

### 3.2. Historical and Ethnographic Sources for the Economic and Social Value of Cattle

Prior to the late seventh century, coinage was not widely used in England, and the wealth and status of a household was denoted, at least in part, by the number of cattle it owned [24], [62] (p. 44). This can be observed linguistically in the Old English term for money *feoh* (fee), from the Germanic *fehu* and Old Norse *fe*, meaning cattle, property, and money; later English terms *capital* and *chattel* and Welsh *da* (good) also derive from the word for ‘cattle’ [62] (p. 45), [63] (p. 59). The use of cattle as currency to pay tributes and fines is described in the *Laws of Ine* (c. AD 694), where the standard food-rent for 10 hides includes, amongst other things, two full-grown cows [60] (document 32.2.2.70). Other documents indicate the importance of cattle to the economy by detailing laws relating to the theft, hire of, damage to and by cattle, and their use [24] (p. 211), [60], [64] (p. 515), [65,66]. It is therefore no coincidence that cattle are the most commonly recorded livestock in zooarchaeological assemblages of this period [10,11].

Coinage became widespread in eastern England from the eighth century, although payment of goods and services continued to be made in kind [67]. While cattle continued to have intrinsic economic value, they were increasingly commodified. In the early tenth century, the laws of King Athelstan valued a sheep at five pence, a pig ten pence, a cow twenty pence, and an ox thirty [60] (document 37.2.2.6.2). A plough team of eight oxen was priced at one pound, the same as a slave [60] (document 141.2.5). This is reflected in the continuing, if diminishing, dominance of cattle over sheep and pigs in the archaeological record until the eleventh century [10,11]. The importance of cattle to the medieval economy is summarized by Walter of Henley in the thirteenth century, who recognized that, “the one necessity was labor; from the estate which was well stocked with men and with oxen a fair income could be derived; but if there was no labor available, the estate could only have a prairie value” [68] (p. 15). A similar sentiment is shared by Algerian farmers, who have a proverb that states, “Wealth comes from ploughing or inheritance” [69] (p. 35), and the early pastoralists of India describe a very wealthy man as, “Lord of cows” [70] (p. 66). By the late fourteenth century, a manorial document from Harton, South Shields prices oxen at twelve shillings a head, a cow ten shillings, and a sheep at two shillings, three pence [71] (document 565. 4.1).

Descriptions of medieval husbandry practices by Walter of Henley are from the viewpoint of an elite landowner rather than a farm manager, but are nevertheless useful indicators of how animals were perceived in the late thirteenth century. It is acknowledged that cattle that were expected to work hard should be fed well, kept clean, not overworked, and that they should be well looked after by a knowledgeable ploughman, cowherd, or waggoner [68] (p. 22). Walter of Henley’s objective was to maximize the profits realized by each manor, and the animals therein were regarded as commodities, but he also realized that it was vital to treat animals well for optimal performance. Specifically, he notes: “if the ox is to be in a condition to do his work, then it is necessary that he should have at least three sheaves and a half of oats in the week” [68] (p. 12).

If cattle were an outward sign of a household’s wealth, it might be expected that pride was taken in the appearance and health of those animals as a reflection of their skill. This can be observed even in modern farming communities. Farmers in Cumbria, England “speak of pride and satisfaction in relation to the process of rearing healthy stock” [50] (p. 104), while a cowherder in Lombardy, Italy also had a “sense of pride and a sense of purpose in his and his family’s success as breeders and the beauty of their herd” [72] (p. 50). The treatment of livestock as individuals, with pride and empathy is not unique to those working with cattle, and similar observations have been made of farmers as diverse as English pig stockmen [73] (p. 64) and Maasi sheep and goat herders in Kenya [74].

Much of the medieval documentary evidence was written for or by the elite, who were the biggest landholders. They could afford to rest their animals, potentially having spare animals, and could afford replacement stock. Indeed, a survey of the manor of Stukeley in the early twelfth century lists three ploughs with 30 oxen [75] (part 4, ch. 1); if one plough was pulled by between two and eight oxen [8] (p. 51), this would allow at least six spare animals to be rested or used for other draught duties. However, they would also be expected to plough more land, working for a greater part of the year than the cattle of peasant farmers, even when cattle had to be shared between households [8] (p. 53), [56] (p. 74), [76]. Given the growing social hierarchy throughout the period, some farmers would not have been able to afford the luxury of high-quality fodder or optimal rest periods for their cattle. Ethnographic descriptions of draught cattle used in modern Nigeria and Asia note that the high cost of replacing animals leads to them continuing to be worked even when they are ill, lame, or severely injured [46,47]. In *Pierce the Ploughman’s Crede*, a poem written in the fourteenth century, the ploughman, his family, and his animals are described in a terrible state:
“I saw a simple man hanging on a plow. His ragged coat was made of coarse material and his hood was full of holes so that his hair stuck out. His shoes were thickly patched and his toes stuck out as he worked. His stockings hung over the back of his shoes on all sides, and he was spattered with mud as he followed the plow. His mittens were made of rags and the fingers were worn out and covered with mud. He sank in the fen almost to his ankles as he drove four feeble oxen that were so pitiful their ribs could be counted. His wife walked with him, carrying a long goad. Her short coat was torn, and she was wrapped in a winding [winnowing] sheet for protection from the weather. Blood flowed on the ice from her bare feet.”[77] (lines 421–436)

The practice of hiring animals is described as early as the late seventh century in the *Laws of Ine* [60] (document 32.2.2.60). For a peasant to be able to offer oxen for sharing is likely to have placed them in some standing in the community, as is the case in modern-day Nigeria [45] (p. 173), and in the villages of Kabylia, Algiers, there is a strong solidarity between those who own yokes of oxen [69] (p. 26). The status of the ploughman is further exemplified in Aelfric’s *Colloquy* written in tenth century England, where the master asks, “And which among the secular arts seems to you to hold the first place?”, to which the counsellor replies, “Agriculture, because the ploughman feeds us all” [78] (p. 113).

The social value of early heavy plough technology that enabled increased production of previously uncultivated heavy, clay soils, is implied by the ritual deposition of implements such as coulters throughout Northwest Europe in the medieval period [79,80]. Some recognition of the importance of cattle within a community may also be evident in the deliberate burial of the lower legs of an animal in a mid-tenth to mid-eleventh century boundary ditch at Ketton, Northamptonshire [81]. The animal had a mPI of 0.13, indicating that it was used for low-level draught purposes as well as possessing age-correlated pathological and sub-pathological changes [15].

Cattle theft was a serious crime referenced in ninth to tenth century documents, and described in the treaty of King Aethelred II with the leaders of the Viking Army (AD 991–994) as an offence comparable in severity to homicide: “if anyone charges a man of our country that he stole cattle or slew a man, and the charge is brought by one viking and one man of this country, he is then to be entitled to no denial” [60] (document 42.2.2.7). Further laws relate to raising the alarm, tracking stolen animals, and paying fines. In an effort to prevent cattle theft, two witnesses were required to confirm ownership when an animal had changed hands [60] (document 34.2.2), and a further ‘two trusty men’ were required in the law code of King Aethelred II (AD 978–1008) for the same reason at its slaughter [60] (document 43.2.2.9.1).

### 3.3. Historical and Ethnographic Sources for Human–Cattle Relationships

Although cattle would ultimately be killed for meat, their constant presence means that there would have been some degree of familiarity between the farmer and livestock. The very act of helping to birth an animal, rear and train it, and work alongside it would provide opportunity for a relationship to develop. Even in modern farming, particularly with breeding or dairy cows that have a long productive life, or orphaned calves raised as pets, such bonds are evident [50] (p. 104), [82] (p. 181). This may not apply to all animals, but certainly to one or two individuals in a herd that exhibited strong personalities [48] (p. 126), [82] (p. 141). Conversely, evidence exists for the opposite to be true, whereby animals that are not in close contact with people (such as beef cattle or hill sheep) are more likely to be treated as commodities rather than individuals. Examples of this have been described zooarchaeologically for horses [83] and ethnographically for cattle and sheep [48] (p. 59), [50] (Figure 2n from reference [50]), [82] (chapter 7).

Stronger emotional bonds may therefore be expected to form between farmers and individual cattle when herds contain animals that are older and work closely with people, either in relation to dairy products or draught work [82] (p. 131). Walter of Henley recommended that the cowherd and oxherd should be familiar with their animals, keep guard, and sleep with them each night [68]. In the thirteenth-century book *De proprietatibus rerum,* Bartholomaeus Anglicus is told by an oxherd that he “pleaseth them [oxen] with whistling and with song, to make them bear the yoke with the better will for liking of melody of the voice” [84] (chapter 7). An interviewee in Fijn’s etho-ethnography of Mongolian herders, when talking about the sheep, goats, and cattle says, “…but I like cows the most because I milk them” [16] (p. 153), implying a further special relationship that develops during the milking process.

Cattle are also portrayed as sensitive, one bestiary describing draught oxen as showing kindness to the other animals that they plough with, further suggesting that they know when a storm is near and are reluctant to leave their stables [43] (chapter 39). The pampas cowboys of Brazil come to understand the body language and personality of the cattle they work with, and the use of violence on animals is frowned upon, rather they use their own body language to communicate and interact with them [85] (p. 121).

Lasting memories of cattle may occur if they represent a defining event. The Maasi of Kenya, for example, give a cow to each boy, and the significance of this event means that every man interviewed, even one of eighty years, could describe his first cow [86] (p. 466). At the other end of the life course, in England, several medieval field-names relate to the deaths of oxen such as *Thertheoxlaydede* in Northall, Berkshire from the thirteenth century, and *Godwynesoxe morieabutur* (where Godwin’s ox died) from Great Bowden, Leicestershire recorded in the fourteenth century [87] (p. 90), [88] (p. 211). In a survey of field-names, Greatorex notes that those depicting places where animals have died uniquely refer to oxen, which reflects “the crucial role played by oxen in agriculture” [89] (p. 18–19), and further implies the social, emotional, and economic importance of cattle in the lives of those who worked the land.

The tension between enjoying a close working relationship with cattle over several years, but then ultimately selling them for meat or even consuming them, can be allayed by enjoying the relationship in the present, without thinking about the inevitable end [26] (p. 181), and by knowing they were well treated in life [82] (p. 143). The Suri of Northeastern Africa do not consider their animals in sentimental terms, despite a considerable proportion of the life of herders being dedicated to the care of their cattle. Rather, they respect them as individuals, to be looked after well but not anthropomorphized [90] (p. 360). Similarly, the Kazakh Mongols do not feel remorse when an animal is killed, as to do so would disrespect the value of the life it lived as an individual [16] (p. 226).

The effect of gender on human–cattle relationships is also pertinent. In many ethnographic studies of cattle cultures, it is men who tend to cattle, herd them, and work them [90,91]. Oxen are also preferred over cows for draught purposes [51] (p. 14), and in the medieval documentary evidence, oxen are more valuable. The emphasis on oxen for draught use may have been the case on demesne estates, yet for smaller peasant farmers, it is possible that cows were put to work [51] (p. 15). Similar scenarios, where male cattle are preferred for traction but cows are used when necessary are described by farmers in Nigeria [45] (p. 173) and elsewhere in Africa [46] (p. 29).

Pictorial evidence of medieval ploughing invariably depicts men working alongside cattle and *Aelfric’s Colloquy* refers to a male ploughman [78] (p. 108). Skeletal evidence also implies that it was men who carried out the heavier agricultural work [92] (p. 699). However, peasant women could also be landowners, and although they were more likely to be employed in milking cows or in the dairy, there was no strict division of labour between men and women [93] (p. 145). In *Pierce the Ploughman’s Crede*, the wife is working beside the plough [77] and in the *Laws of Ine*, a widow was to be provided with a ‘cow in summer, an ox in winter’ [60] (document 32.2.2.32), with the implication that she could use the oxen for ploughing her own land or for hiring out. Although by no means universal, ethnographic studies have identified broad, cross-cultural, gender-defined roles whereby men are more likely to herd animals away from the settlement, while women are commonly to be found working with livestock close to the domestic sphere, particularly milking [94] (p. 29). It is therefore likely that in medieval England, not only were the roles of people defined by gender, but also those of cattle. In her exploration of modern English farming, Wilkie also observed that women are also more likely to be emotionally sensitive to the needs of livestock [82] (p. 54), which was described by more than one interviewee as being the result of their own maternal experiences. It might therefore be suggested that the dairy cattle and calves kept close to the house were more likely to have close relationships with women, while the draught oxen working away from the domestic area would be closer to male herders.

There are three developments that may have been detrimental to the human–cattle relationship. The first was the view, spread through Christian scripture, that animals were created to serve humans, the absence of an animal soul meaning that people were not obliged to feel remorse for their suffering, thereby providing a means of exploiting animals while remaining impassive [95] (p. 22), [96] (p. 203). Second, from the tenth century, the increased need to provide regular surplus to pay rents would have led to greater pressure on those working the land to increase production and work cattle more intensively [95] (p. 23), [97] (p. 183). Even pastoral herders such as the Maasai of East Africa, who moved to commercial production in recent times, have lost their traditional relationship with cattle and their ability to recognize and describe their animals as individuals [98] (p. 210). Third, social inequality became more pronounced between the eleventh and thirteenth centuries [9,99]. The hierarchy in place by the thirteenth century meant that even free men were tied to their lord through compulsory labour [100] (p. 33), and the majority of free peasant farmers had to subsidize the production of their own land with a waged income in order to live [99] (p. 295). The Domesday survey of 1086 suggests that as much as three quarters of the wealth recovered from the land (e.g., in grain and wool) was in the hands of a small number of elite landholders who ran the royal estates [101] (p. 16).

Increasing social hierarchies based on inequality are evident during the course of the medieval period in England and the relationship between cattle and humans was also inevitably based on disparity. However, the agency of cattle to affect how they are treated should not be underestimated. Armstrong Oma presents this as a social contract between farmer and animal [26] (p. 178), which in the medieval period took the form of responsibility for feed, care, and shelter on the side of the farmer, and the provision of power, milk, and calves by cattle. Although ultimate control is held by humans, the interdependence of both parties means that for it to be a successful contract, respect would be required by both, “humans and animals are engaged in mutual decision-making… the agency of animals means they are a doing or a becoming, formed by social interactions” [26] (p. 179). This may be applied to medieval farming in relation to the willingness of oxen to plough and cows to let down milk being dependent on the plough hand or dairy worker’s daily evaluation of conditions, for example, whether it is too wet to plough a certain field, or if it is too soon after calving to expect an animal to provide enough milk without being to the detriment of the calf. Thus, although the human is the dominant partner, the relationship relies on mutual trust. While it could be argued that animals can be forced to do their jobs by cruelty, a mutual relationship is necessary to maximize the outcomes [26] (p. 181), which is evident in the recommendations by Walter of Henley to provide good, even lavish, care of working cattle.

One final historical source summarises this relationship between human and cattle. Paulinus of Nola writing in Gaul in ca. AD 400 describes a peasant who rented out the two oxen he owned. These cattle were prized above even his own children, but they were stolen. In his grief, he prayed at the local shrine of St Felix, “…then went home in the dark to lay inconsolably in the filth of the oxen’s empty stall, caressing their hoofprints”; observing his deep sorrow, the saint returned the oxen, and upon their return, “the oxen and peasant embraced one another: they gently nuzzled their kindly lord and fawningly caressed his breast in turn. The horns of his beloved cattle did him no injury; he drew their heads as though they were soft to his proffered breast” [102] (p. 234). This illustrates both the social and economic value of the cattle (the gains from renting them out, his love for them), but also the deep, reciprocal affection of both parties exemplified by their joy at being reunited.

### 3.4. Integration: Towards an Account of Changing Human–Cattle Relationships

The intrinsic value of cattle as a store of wealth is a common theme in many ethnographic studies and can be observed linguistically in the pre-coinage society of the fifth to sixth centuries in England. This is evidenced in the zooarchaeological record by the dominance of cattle over other livestock between AD 450 and AD 650, reflecting their importance as portable wealth and their role in transactions (such as the payment of fines, tributes, and dowries). Later, when cash became more important to the economy, cattle continued to be the most valuable of the major livestock species, second only to horses. Although there is less direct evidence for the pride taken in the health and appearance of cattle by farmers, it is reasonable to infer that this was so from the historical and ethnographic record. Although until the mid-ninth century cattle were not often kept until any great age, ethnographic narratives suggest that cattle that embody wealth are commonly treated with respect, as individuals, and it may thus be expected that during this period there was a positive human–cattle relationship based on the intrinsic value of one partner, and the pride taken in owning and caring for them by the other.

Repeated, cross-cultural findings exist for sympathetic and affectionate relationships between people and the cattle with which they have close working relationships such as plough teams and dairy cows. Such relationships are likely to have been increasingly common from the mid-seventh century, but particularly from the mid-ninth century, when the zooarchaeological evidence attests to the increased use of cattle for secondary products, specifically draught use. Historically, this is evident in an increasing population, a greater proportion of which occupied urban centres requiring a supply of grain from the rural hinterland [9] (chapter 2).

Much recent ethnographic and social zooarchaeological discourse has been concerned with eschewing the anthropocentrism of Western thought, where humans were believed to have dominance over nature. However, the re-introduction of Christian doctrine in the seventh century in England inevitably led to the spread of this dichotomy. This change in emphasis would have necessitated a shift in the human–cattle relationship that would have been complex and conflicting. On one level, those who lived and worked with cattle, both draught and dairy animals, likely formed close relationships with their cattle, would recognize animals as individuals and even acknowledge the social contract between cattle and farmer. However, by the tenth century, the commoditization of cattle combined with the need for greater agricultural output, may have been a contributory factor to the increasing influence of church doctrine emphasizing the superiority of humans over other animals, thereby unbalancing this mutual respect. The view spread by the church that animals were provided by God for the benefit of people would have made the necessary adaptations to production required to supply a rapidly increasing population, growing urban market, and international trade network with food and raw materials, morally acceptable.

From the mid-eleventh century, draught cattle continued to be vital to the economy, yet the increase in beef production most likely reflects a consumer demand created by a thriving, fully urban population and the ability of peasant farmers to supply the market [9] (p. 164). Ethnographic examples suggest that beef cattle, which would only be alive for two or three years, would have less time to form bonds with the farmer. With less chance for mutual respect to form between human and animal, this strategy, where stock was treated as a commodity, would also have eased the transition between living animal and beef carcass.

It is therefore feasible that in medieval English society, those who lived and worked with dairy cattle and plough oxen would have been aware of the animals as individuals, building relationships based on trust and familiarity, despite the knowledge that they would ultimately be killed for consumption. In turn, cattle provided wealth and status, and a means of providing food and power vital to the economy. Their ability to form relationships based on trust and familiarity would also have been recognized by the people working with them. The increasing development of markets and urban populations meant that that for many people, the bond with livestock was broken, making it easier to commodify animal carcasses and demand increasingly young animals as food.

## 4. Conclusions

Zooarchaeologists have generally been reluctant to use animal bone evidence to interpret the dynamics of relationships between people and animals in the past, given the potential for accusations of conjecture (though see [103]). This exploration of human–cattle relationships has sought to overcome such charges by combining the evidence from animal remains with an in-depth exploration of historical documents to provide a backdrop of attitudes towards animals, in combination with cross-cultural, repeated observations of people working closely with cattle in a variety of situations.

Humans are social animals, and their ability to form relationships with other social animals such as cattle should be integrated into archaeological narratives as much as their ability to make pots and use currency. The difficulty lies in the intangible nature of such relationships, and it is hoped that this paper presents a first step towards a deeper consideration of the ways that people interacted with the natural world in the past.

## Figures and Tables

**Figure 1 animals-11-01174-f001:**
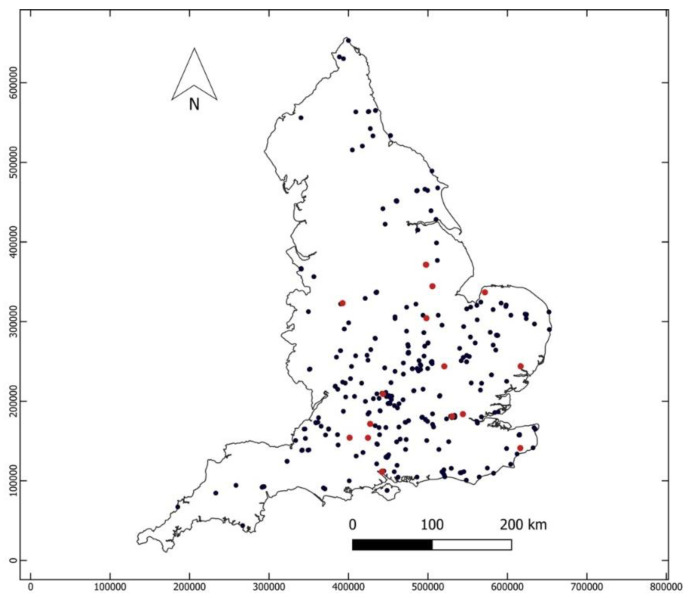
Location of all sites included in the dataset. Full details can be found from the Archaeological Data Service [33]. Targeted sites are shown in red.

**Figure 2 animals-11-01174-f002:**
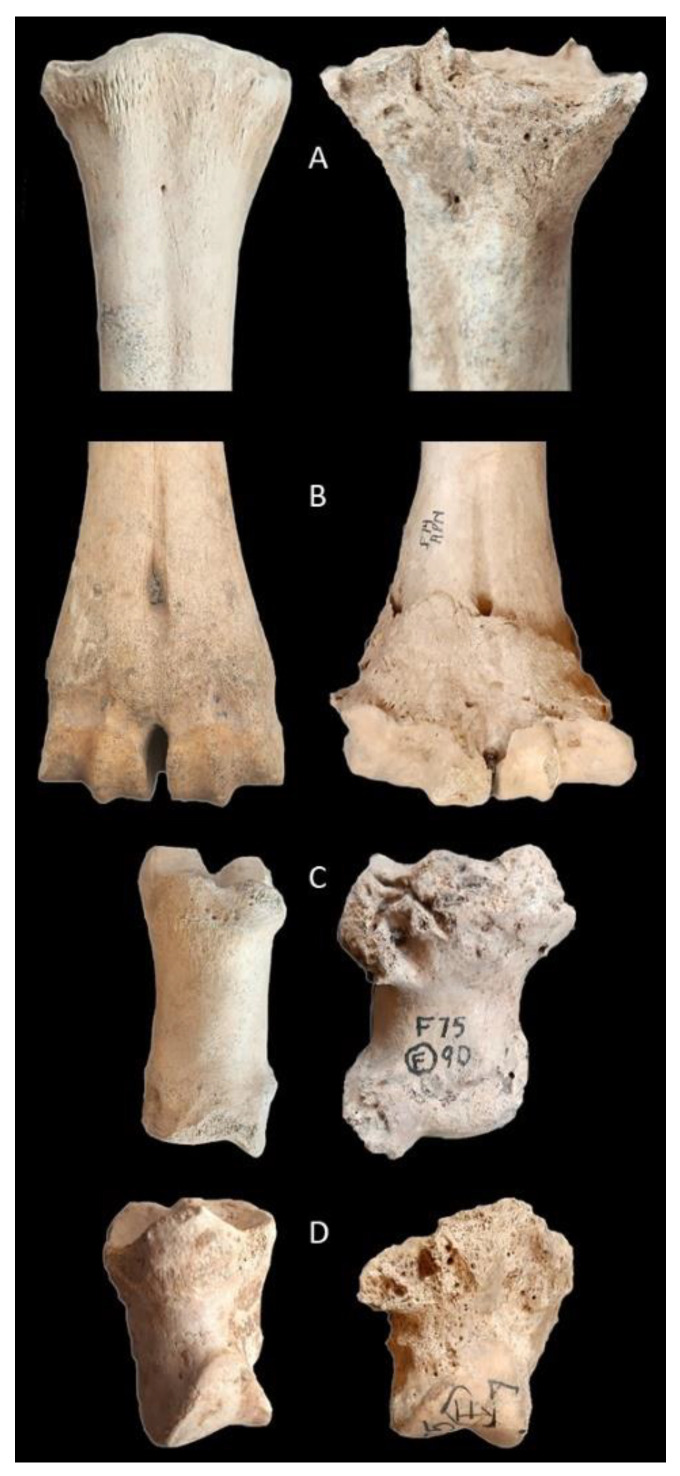
Selected cattle foot bones (anterior view) illustrating examples of severe changes. ‘Normal’ examples are shown on the left and lesions on the right: (**A**) proximal metatarsal illustrating stage 3 lipping and exostosis; (**B**) cattle metacarpal, showing stage 4 exostosis and broadening; (**C**) first phalanx showing stage 4 proximal and distal exostoses; and (**D**) second phalanx illustrating stage 4 proximal lipping and exostosis.

**Figure 3 animals-11-01174-f003:**
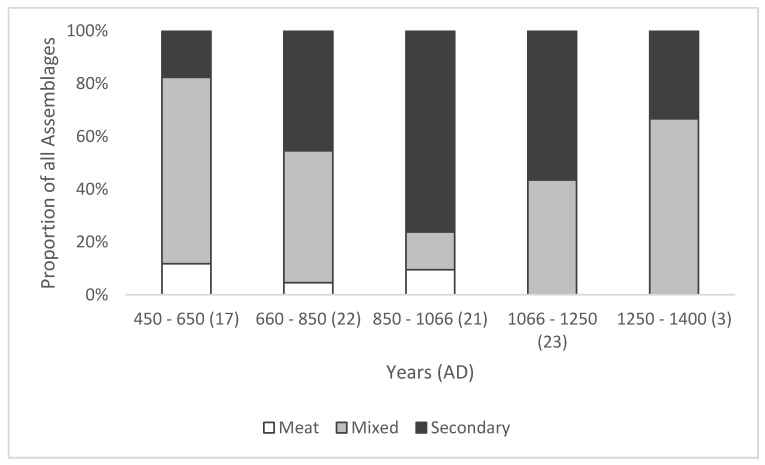
Assemblages representing cattle culled for meat (mostly younger animals culled prior to or at the age of skeletal maturity), following their use for secondary products such as milk or traction (mostly older adult animals that are fully skeletally mature), and those that were split between younger and older animals. Data from all available sites in the database. (*n*) = number of assemblages.

**Figure 4 animals-11-01174-f004:**
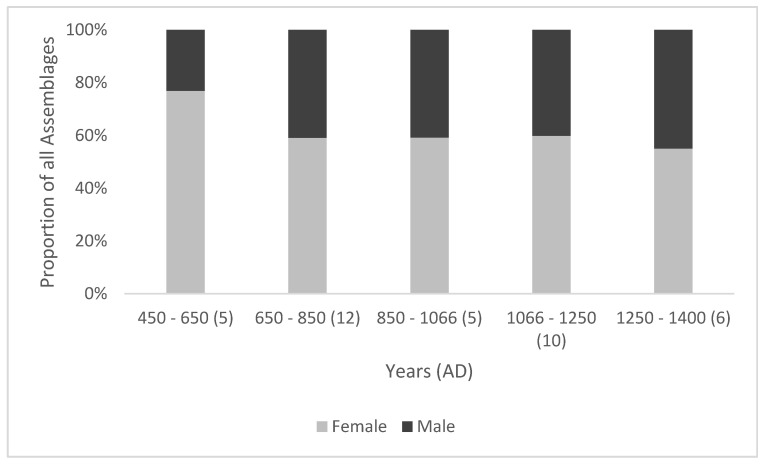
Relative proportion of male and female cattle pelves recorded from the targeted sites. (*n*) = number of assemblages.

**Figure 5 animals-11-01174-f005:**
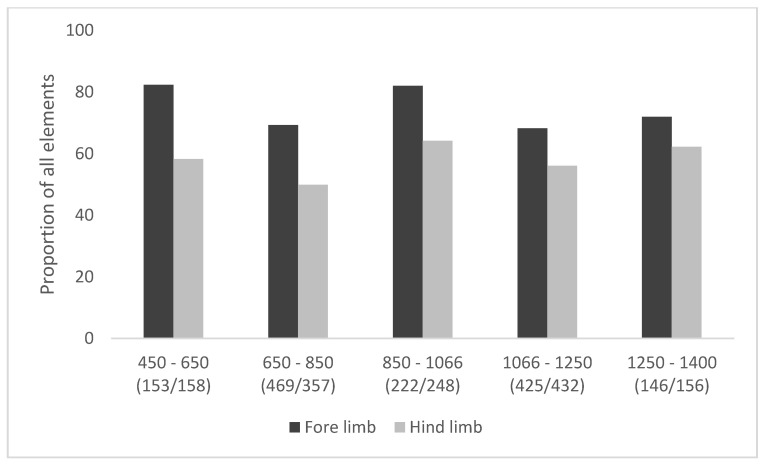
Cattle foot pathology data showing prevalence (proportion of all elements affected) through time. Data from targeted sites. (*n*/*n*) = number of elements from the fore and hind limbs.

**Figure 6 animals-11-01174-f006:**
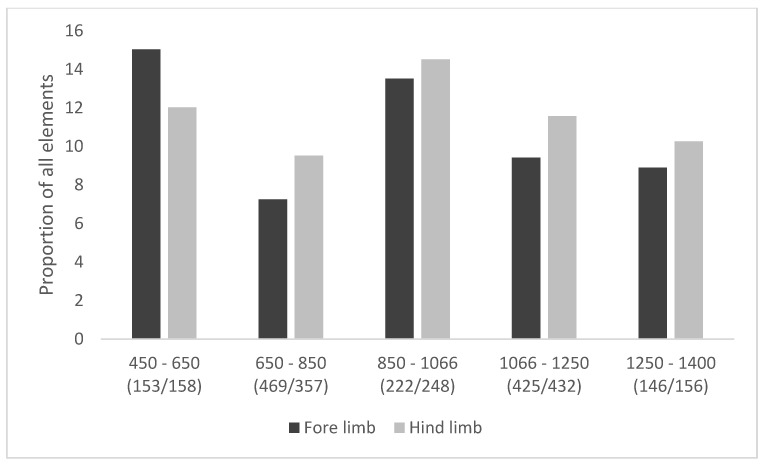
Cattle foot pathology data showing severity (proportion of all elements with a score of >0.40) through time. Data from targeted sites. (*n*/*n*) = number of elements from the fore and hind-limbs.

**Table 1 animals-11-01174-t001:** Summary of all sites included in the dataset by phase and site type.

Site Type	AD 450–650	AD 650–850	AD 850–1066	AD 1066–1250	AD 1250–1400
Ecclesiastical		6	5	14	6
High-Status	2	9	9	52	18
Rural	53	36	15	32	17
Urban	5	30	83	129	61
Total	60	81	112	227	102

**Table 2 animals-11-01174-t002:** Targeted sites showing the phases of occupation and the number of cattle foot bones (metapodials and phalanges) recorded.

Site	County	Phase (Years AD)	No. Bones
Barking Abbey	London	500–850	2
		675–850	3
		850–1066	1
		1066–1200	12
		1200–1400	3
Bow Street	London	600–750	54
Collingbourne	Wiltshire	700–900	13
Cook Street	Southampton	650–875	111
Eynsham	Oxfordshire	500–650	14
		650–850	144
		850–1066	12
		1066–1300	187
		1200–1330	44
Flaxengate	Lincoln	870–1090	184
		1060–1200	224
		1200–1400	73
French Quarter	Southampton	900–1066	99
		1066–1250	105
		1250–1350	107
Ketton	Northamptonshire	850–1066	11
Lyminge	Kent	400–700	86
		600–850	142
		1100–1300	15
Market Lavington	Wiltshire	400–700	163
		700–900	2
		900–1175	0
		1100–1300	5
		1300–1400	2
Quarrington	Lincolnshire	450–650	41
		650–900	37
Ramsbury	Wiltshire	750–850	45
		800–1300	3
Sedgeford	Norfolk	650–875	43
		800–1025	95
Stafford	Staffordshire	900–1100	2
		1100–1300	46
Stoke Quay	Suffolk	700–875	163
		825–1100	55
		1050–1200	89
		1150–1400	42
Stratton	Bedfordshire	400–600	16
		600–850	42
		850–1150	54
		1150–1350	31
West Parade	Lincoln	1050–1300	144
		1275–1375	35
Total			2801

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
