# Peer review of "Close Companions? A Zooarchaeological Study of the Human–Cattle Relationship in Medieval England"

_animals, 2021, doi:10.3390/ani11041174_

Round 1
Reviewer 1 Report
The manuscript took in account most of the comments and is now very well written and interesting because it is enlarged to a more suitable/clarified context.
Please, to hide/blind my identity.
Author Response
Thank you for your helpful comments
Reviewer 2 Report
The paper is well structured and the goals introduced at the beginning are reached and clearly presented. I strongly recommend the publication of this interesting article.
Author Response
Thank you for your review
Reviewer 3 Report
This article is a welcome addition to the current trend to expand interpretations of zooarchaeological data with the addition of historical data and anthropological analogues. Here, the authors attempt to contextualise slight temporal shifts in cattle husbandry and pathological incidence across medieval England with historical records and worldwide ethnographic examples.
The hypothesis suggested by the authors to account for zooarchaeological changes outlines a shift towards urban market economies as town centres proliferated in the medieval period. However, this article could benefit from a more nuanced and detailed discussion that connects the historical and ethnographic examples to a more detailed discussion of zooarchaeological trends. For example, the authors suggest that the ‘social bond’ between humans and cattle decreased with increasing urbanisation, resulting in a change of the status of cattle from essential partners in agricultural production to disposable commodities. However, they fail to explicitly point out historical data that would support this change in attitude.
Slight rhetorical changes in the discussion section of the MS would greatly strengthen the arguments laid out in this article. As it stands, the zooarchaeological data and the historical/ethnographic section are disjointed and the historical context is not explicitly or elegantly supporting interpretations of the zooarchaeological data.
Furthermore, some other topics that the authors raise in passing might also be touched on in the discussion in more detail, such as the idea that agricultural exploitation of regions with underlying clay geologies might have cattle with a higher incidence of pathologies, or that a site type-based comparison through time (e.g. rural v urban) of pathological rates might control for any recovery bias caused by the proportion of rural and urban sites excavated through different time periods. However, at this stage, the article will be acceptable for publication after minor rhetorical adjustments as a broad stroke piece about the state of cattle in medieval England, and will be a useful jumping off point for others interested in this topic.
